# Overuse of Health Care in the Emergency Services in Chile

**DOI:** 10.3390/ijerph18063082

**Published:** 2021-03-17

**Authors:** Ximena Alvial, Alejandra Rojas, Raúl Carrasco, Claudia Durán, Christian Fernández-Campusano

**Affiliations:** 1Departamento de Ingeniería Industrial, Universidad de Santiago de Chile, Santiago 9170124, Chile; ximena.alvial@usach.cl; 2Escuela de Obstetricia y Puericultura, Universidad de Santiago de Chile, Santiago 9170022, Chile; alejandra.rojas.r@usach.cl; 3Facultad de Ingeniería, Ciencia y Tecnología, Universidad Bernardo O’Higgins, Santiago 8370993, Chile; 4Facultad de Ingeniería, Universidad Andres Bello, Santiago 7500971, Chile; 5Departamento de Ingeniería Industrial, Universidad Tecnologica Metropolitana, Santiago 7800002, Chile; c.durans@utem.cl; 6Departamento de Ingenierías Multidisciplinares, Universidad de Santiago de Chile, Santiago 9170022, Chile; christian.fernandez@usach.cl

**Keywords:** Chilean health system, emergency service, triage

## Abstract

The Public Health Service in Chile consists of different levels of complexity and coverage depending on the severity and degree of specialization of the pathology to be treated. From primary to tertiary care, tertiary care is highly complex and has low coverage. This work focuses on an analysis of the public health system with emphasis on the healthcare network and tertiary care, whose objectives are designed to respond to the needs of each patient. A review of the literature and a field study of the problem of studying the perception of internal and external users is presented. This study intends to be a contribution in the detection of opportunities for the relevant actors and the processes involved through the performance of Triage. The main causes and limitations of the excessive use of emergency services in Chile are analyzed and concrete proposals are generated aiming to benefit clinical care in emergency services. Finally, improvements related to management are proposed and the main aspects are determined to improve decision-making in hospitals, which could be a contribution to public health policies.

## 1. Introduction

The public health system in Chile is made up of care facilities that serve the entire population registered in this modality, approximately 80% of the inhabitants; according to the seriousness of the patient’s condition, without any distinction. It is managed by public entities who depend on the Chilean Ministry of Health (MinSal), with the system being regulated by Law N${}^{\circ }$ 19.966 of universal access with explicit guarantees and by Law N${}^{\circ }$ 19.937 which updates the administration of the health authority. It has three basic functions, including: (1) funding management, (2) regulation, supervision and control, and (3) the provision of the public health benefits. The first function is conditioned by National Health Fund (FONASA) which manages the funding, administration of the benefits and health coverage for people along the country. The second function is related to the establishment and supervision of health policies, standards, plans, and programs through the MinSal and its entities. The last function is based on the execution of the public health programs and the management of the system itself [1].

The reality of doctors of the public sector is an important concern, since they must care for a great number of inhabitants, unlike those professionals who work in the private sector which provides an independent service that can only be used by those who have better economic conditions to face the payment of the medical attention. This is shown in the study generated by the Undersecretary of Assistance Networks and the World Bank where it is noted that in 2011 the number of physicians per 1000 inhabitants was of 1.58, while in 2018 this index increased only to 2.62 [2]. It must be highlighted that, considering the total number of doctors, 56% of them care for 17% of the inhabitants in the private sector, while 44% of them care for 80% of the population in the public sector. The remaining 3% of patients of the private sector depends on the Health Services covered by the Armed Forces. In the same sense, according to the Organisation for Economic Co-operation and
Development (OECD) in Chile in 2016 the medical proportion per capita was 2.3 physicians per 1000 inhabitants, while for the OECD in the year 2017 it was of 3.5 physicians per 1000 inhabitants, and at present this low number has not changed significantly. Even two years later this index did not vary significantly [3]. As an attempt to solve this critical situation, it is required to study the causes and the difficulties generated in the patient’s service experience more profoundly [4].

This article will focus on the public system of the healthcare network of the Chilean Health System and the tertiary care, whose objectives are designed to respond to the needs of each patient [5]. The research includes hospitals, self-management establishments, lower complexity facilities and the emergency network of high complexity and reduced coverage in which the admission is made through medical interconsultation derived from the primary and/or secondary care and/or a direct admission, and by direct admission of an emergency category depending on the seriousness of the patient condition [1]. In the same line, the Emergency Services will be analyzed, whose objective is to provide rapid medical attention according to the user’s demands in relation to their serious condition [6].

The studied public health service has a problem that affects the system directly, which is related to the overuse of the emergency service. This situation originates when patients who should be taken care of by other levels of attention congregate in the emergency rooms producing a saturation of medical attention. This is why there is a need for a definition of the criteria to determine when there is a medical emergency, when this corresponds to a clinical situation that would probably generate a deterioration, risk or danger for the patient’s life, in relation to the time that takes place between its occurrence and the implementation of an effective treatment [7,8]. That is to say, those patients whose emergency and serious condition cannot wait for a regular conduit, since they need to be treated immediately because their life is in danger. Therefore, the emergency comes conditioned by the time that passes until the definitive care of a health professional, in this case, the doctor on the current shift, and for the availability of the health network. The seriousness of each condition is related with the diagnosis given, which starts from a first evaluation of the medical staff conditioned by the TRIAGE that classifies the patients to make an orderly attention. For example, a cancer is more serious than an asthma crisis, but the latter can be more urgent than the first one.

The paper is distributed as follows: Section 2 presents the Background. Section 3 presents the Materials and Methods. Section 4 presents the Description of the attention process. Section 5 shows the Results and Discussion. Section 6 shows the Limitations of the study. Finally, Section 7 presents the Conclusions.

## 2. Background

### 2.1. Emergency Service

In relation to the Public Health Service, there are three levels of complexity and coverage depending on the seriousness and degree of specialization of the pathology to be treated: (1) primary care, of minimum complexity and wide coverage that includes outpatients consultations and health problems that can be solved with less resources and without hospitalization; (2) secondary care, of intermediate complexity and medium coverage, that assists more complex procedures or the type of outpatient surgery and (3) tertiary care, of high complexity and reduced coverage, in which there are considered patients who require specialized procedures and cutting-edge health technology.

As is shown in Table 1 in 2017 at a country level, 19,382,057 emergency attentions were presented representing a 43.4% of the total, wherein 17,009,660 corresponded to primary care and 8,238,964 to the specialty care.

Figure 1a,b show a heat map where it is observed that there is a lack of health workers per 10,000 inhabitants, a critical situation that is one of the causes of the overuse of the emergency service.

The lack of doctors and other health professionals is critical. Despite the efforts made by the Department of Studies, Planning and Control of Human Resources Management of the MinSal, the problem has not yet been solved [9]. Several reasons have been reported by the Undersecretary of Assistance Networks of the Ministry of Health, and the following stand out: immigration has increased in the last decade [10,11], the changes and improvements that have been generated in the epidemiological standards, and the shortage of professionals in the area of mental health, emergency, intensive care in adults and children, family and anesthesiology [12]. One of the measures that have been implemented is the hiring of foreign doctors but this has not been effective due to the high failure rate that exists when they take the Single National Knowledge Exam [13]. On the other hand, there are greater incentives to work in the Private Health System than in the Public System [14].

### 2.2. The Health Service User

The health service must be analyzed systematically since the users are the patients and the service providers; even though the patient is the one that suffers the problem directly. For the whole understanding of the health network, the party providing the service cannot be forgotten, which in one way or another is also affected by the overuse of the attentions in the emergency service [15,16]. Regarding the Primary Care Centers, the main claims handled in the Office of Information, Claims and Suggestions (OICS) is that they present certain shortcomings to efficiently solve their health problems such as: the human resource is scarce, the attention tends to be depersonalized, and the management of medical appointment scheduling and confirmation of hours is mostly less efficient compared to the Private Sector, which has a greater implementation of Information and
Communication Technologies (ICT). In fact, of more than 2000 Primary Care Health Centers, only 42 have been accredited to date [17,18].

The main affected or the recurrent user is the patient, despite paradoxically being the one that starts the root of the problem, mainly for the lack of hours of attention and available resources [19,20]. These patients are characterized by showing some kind of sudden annoyance that disturbs their welfare status. To the above mentioned is added the lack of information or inexperience that they have of the health problems, causing their appreciation of their problem always to be an emergency, therefore making a bad use of the service [6,21]. People at the Emergency Department of the healthcare service have the perception that in one way or another their problems will be solved, even if they are not urgent. This situation with random characteristics makes the generation of statistics and the forecasts of attention by age rank defined for a type of user more recurrent in the consultation [22,23] difficult. Then, a heterogeneous pattern is generated.

In relation to the providers of the health service, it can be said that these are mainly medical professionals, specialists, nurses, nursing technicians who are in charge of giving direct care and giving solutions to the problems presented by the patients and become the direct contact that can answer their doubts, as complaints that can arise for the delay in the attention [24]. Receptionists are common people that do not possess professional studies in the health sector, but they are the ones in charge of the admission of the patients. Then, the patients timely receive the medical care from the paramedic technicians, nurses and doctors, who finally perform the evaluation of the general status of each patient.

For the medical evaluation the Triage shown in Table 2 is used, which is defined by the Ministry of Health and corresponds to a categorization method that allows the organization of the medical care for the users, depending on the resources available and the condition of each individual [25]. Triage is a classification that allows quick attention to the more serious patients first and then to the less serious patients. Its importance resides in that this method is used to classify patients in an objective way depending on their risk and seriousness to achieve a correct prioritization of the attentions, and it is used across the entire health network of our country [26].

According to the above, in order to demonstrate the reality of the medical attention of patients in a hospital facility, it is studied the case of the emergency department of Hospital Dr. Sótero del Río in which was used the Triage categorization. It is observed that of the total patients that receive medical care, less than 1% of the users correspond to patients categorized in C1 (around 0.8–0.9%); most of the patients are categorized as C2 and C3, being the main focus of attention (closet o a 40%); while the less serious cases are categorized in C4 and C5, corresponding to 60% of the visits to the health service when they could be perfectly cared for at the primary and/or secondary attention level, according to their categorization of seriousness.

Therefore, it can be said that the Triage stage includes the evaluation of the vital signs and it is determined as one of the critical processes within the attention that forms a bottle neck in the distribution of the demand inside the emergency room [27,28].

## 3. Materials and Methods

This work studies the excessive use of Emergency Services in the Chilean Health System in the Public Attention Network. To investigate the possible causes and generate solution proposals for the process, a cross-sectional descriptive design was carried out in which field data was collected on the users’ perception of the waiting time and the degree of satisfaction they had with the received medical care. Surveys with open and closed questions were applied to 101 people between June and August 2016. The sample consisted on the general public, patients who attended the Emergency Department to check health problems and health professionals with experience in the Metropolitan Region such as doctors, nurses and midwives, among others.

It should be noted that cases of three medical facilities were analyzed, due to their high complexity and greater coverage in Chile. These are the South East Metropolitan Health Service (Hospital Sótero del Río), Central Metropolitan Health Service (Hospital El Carmen de Maipú-Dr Luis Valentín Ferrada) and the South Metropolitan Health Service (Barros Luco Trudeau Hospital).

On the other hand, to identify the design of the medical care process and identify the causes of its problems, an analysis of official and statistical documents of the government agencies of Health Statistics and Information Department, the MinSal and OICS was carried out between the years 2016 and 2020 [29]. The aforementioned repositories contain national information and include the following data: volume of emergency care in the Chilean Public Health System ordered according to the timeline, statistics of claims in relation to emergency care received in the Assistance Network (AN) and expected performance for the Emergency Services of the AN.

Subsequently, improvement proposals were prepared based on the 101 surveys that were carried out, the analysis of the processes is detailed in Section 4, the investigations in Emergency Services in fast tracking, and the use of ICT in telemedicine [30,31] for remote access and sharing of patient records and medical records [32].

Finally, in Section 5 the proposals were evaluated by applying surveys with closed questions to 21 experts in Emergency Medicine, including Physicians in clinical practice in Emergency Services, public health doctors, nurses and midwives specialized in emergency medicine. The instrument uses the Likert scale from 1 to 5, wherein 1 = insufficient, 2 = sufficient, 3 = fair, 4 = good, 5 = excellent.

The difficulty in accessing information from local records and statistics that generally receive a very confidential handling as well as the lack of time of the experts due to the excessive workload is mentioned in Section 6 on the limitations during the interventions carried out.

## 4. Description of the Attention Process

### 4.1. Emergency Attention Process

Given the variety of conditions in which patients arrive at the emergency room, preference is given to those of greater severity with a rapid, exhaustive and rigorous first assessment based on the protocols already established or on the professional criteria in charge; considering the current vital risk and the possible future complications they may have [33].

The present work describes the operation of an average Emergency Service, data and statistics. It should be noted that a generic average process is studied since the activities and waiting times of each of the establishments have a high variability due to seasonality, geographical location and hours of operation [34].

The emergency care process comprises the following activities:It begins when a person detects an anomaly that must be solved immediately because it causes malaise, pain, fear or some discomfort.The patient goes to the Healthcare Center, which has an emergency department, including an admission area of the service [35].Patient’s information is registered and a patient file is created.After a few minutes of waiting, they are called for an evaluation according to the severity and the type of health problem.According to the waiting time assigned to each category in the Triage scale, the severity of the patient’s condition is determined as of greater or lesser degree of importance [27,36].An evaluation is made and the patient awaits their attention according to the waiting time established in the Triage.Notice is given to the patient for their healthcare and according to the categorization, they go to an assigned diagnostic room, which could be separated by specialties such as medicine, traumatology, surgery and maxillofacial sectors.The patient is examined by the doctor on duty who asks for a description of the symptoms and requests tests if necessary.The patient awaits the diagnosis or the results of its laboratory or radiology tests.The diagnosis is given to the patient and the doctor indicates the steps to follow for their recovery.The patient leaves the Healthcare Center or is hospitalized for subsequent intervention according to their diagnosis, and the process is completed.

An average care process in Chilean public health has been represented in the flow diagram shown in Figure 2, and the average waiting times are shown in Table 3. It should be noted that these average waiting times are determined based on the compilation of statistical data from the medical care records of four Emergency Services in the Metropolitan Region, carried out during 2019.

It is observed that in activity 2 a saturation of the process is generated as less serious cases accumulate, causing a bottleneck in the performance of the Triage. One of the reasons why this occurs is due to the extremely delayed medical attention, wherein patients must wait several months or even more than a year to be able to make an appointment with a doctor in the public sector. Patients prefer to go to the Emergency Department and wait for the entire process no matter how long it takes to receive medical attention.

On the other hand, the following waiting times are influenced since the calculation of the performance of health personnel in the Emergency Services is not consistent with the number of patients who attend it [37], since the number of consultations frequently exceeds the estimated average [7]. This often causes health personnel to be overworked, causing increased leave due to stress or other mental health conditions [38]. The medical care boxes are not enough, causing patients to wait on stretchers located in corridors of the physical plant once their emergency care has begun, and even pseudo-hospitalizations sometimes occur in corridors while clinical procedures are being performed and/or for quotas to be generated in the corresponding tertiary service.

Of course, this situation negatively impacts both the internal user (health personnel) and the external user’s (the patient) perception of satisfaction [39]; in addition to hindering the normal flow in clinical care, which is less fluid and develops a tendency to less protection of privacy in care. Finally, sometimes these shortcomings cause a delay in the care and/or the resolution of the cases, directly affecting the health of people who must tolerate the suffering of their ailments without an early clinical solution, some palliative measure, or a worsening of their symptoms during the waiting time.

### 4.2. Causes of Overuse of the Emergency Attention

As indicated in the previous point, there are aspects in PHC that affect its adequate performance; also in the Emergency Services, where in the end there is overuse, saturation of the system, and failures in the provision of the health benefit [40].

Among the main causes that have been reported by both internal users (health personnel) and external users (patients), we can mention:Poor coordination between the patients and the health providers. An example of this situation is observed when an extension of the waiting time occurs, generated by the saturation of the facilities because of the excess of demand produced by the people who go to the health facility with non-urgent needs [41].Patients have limited Access to the PHC [42] due to the lack of available medical appointments in the short term.Emergency attention offers more convenient attention hours [43], including weekends [37], and provides immediate solution to the patient’s medical condition.In the primary care the anomaly situations must be searched for, which allows the detection of the pathology in a timely manner and thus, to achieve a correct referral to the Emergency Services for those patients who require other types of care, if necessary.There is not always a medical history of the patients who go to the Emergency Services, which produces delays at the time of their care in a health establishment due to the need of reconstructing their medical records from the beginning. In fact, despite the fact that the patient is registered in the Public Health System and is treated in their corresponding Assistance Network following all its regular channels, there is no online clinical file among the health institutions of the same System.Little education and promotion in the patients in relation to the preventive health measures and the good use of the service in general terms [44,45].Close to 88% of the PHC are of Municipal administration. The others respond to the Ministry of Health [46]. This shows a lack of unification of the service and among administrations.Hospitals have legal obligations according to law N${}^{\circ }$ 20.584, which regulates the rights and duties of the patients in relation to the actions linked to their health attention and finance; and the Emergency Law N${}^{\circ }$ 19,650 which guarantees that people who are in a vital emergency condition be cared for in the healthcare center closest to where they are. Therefore, despite the existence of an abundant flow, they cannot be sent to their homes [47].Lack of incentive for general practitioners and health professionals who are part of the PHC structure to specialize in family medicine [48].Few economic resources earmarked for the hiring of more professionals for the public health network in the three levels of attention. The medical staff must care for more patients than their capacity. For example, it is common that a nurse assists between 3 and 5 boxes, and a doctor assists between 6 and 10 attention boxes [8], producing a sub-optimal attention, a greater wear of the personnel, and an increase in the risk of making mistakes.Greater investments are required for Health Technology Assessment to improve centers, favoring faster and more effective processes to meet the demand of the population [49].

A better management and modernization of the flow of attention and the categorization of the patients according to their different requirements [28] is necessary. There is a lack of completion of timely information to know expeditiously the demand, so the decisions are made based on the experience of the area Director; and a lack of resources to complete the attentions in an effective way in the hours of under pressure medical care.

On the other hand, it is advisable to do a statistical analysis of the reports received by the OICS Network, which classifies the type of claims, their frequency, exposing their evolution over time and how they react based on the injection of resources and/or modifications in the System.

## 5. Results and Discussion

The proposals in Table 3 were generated based on the analysis of processes, the interviews mentioned in Section 3 and the bibliography related to Emergency Services, such as: fast tracking with the use of ICT that is applied in telemedicine [30,31] for remote access and sharing of historical patient records [32] and the most frequent reports that are reported in the Network channels which are received in the OICS Network [50].

As a result, here are some proposals that could be applicable within the different levels of care in which there is an innate interconnection with emergencies, in which patients and service providers interact in response to vital risk:Generate new information channels directly among the health centers or through channels as Social Media, so that patients can decide by themselves which health center to go to in order to solve their ailments; and finally, between both parties, be able to get a closer and more human relationship.Define the attention flows for the different category of patients, like a fast tracking sector or an area of fast attention for C4 and C5 patients, allowing the observation areas to be purely used by patients that can develop changes in their condition, such as C2 and C3, which have more probabilities of worsening or improving their condition. Managing the demand of patients of medium seriousness and those not serious (C3–C4), waiting times can be reduced, avoiding the misuse of the attention or the resources intended for really serious patients in addition to improving the perception of the service [26].Generate incentives for better work environments in resource management, work environment, and equipment so that health professionals are motivated and contribute more to the public health service. In this way, more efficient responses could be delivered to the population.Encourage the training of the professionals of the primary attention through the specialization in family medicine, with the goal of giving more accurate diagnosis and educating the population in relation to their health condition.Promote the education in emergency medicine for those professionals that are dedicated to this type of attention, so they can give faster and better clinical responses [51].Use the ICT to create systems that allow the obtention of automatic and online medical records of the patients [52]. The information could be displayed in a digital tab which could be seen in any kind of facility where the patient asks for attention [53].Generate more coordination mechanisms among the three levels of primary, secondary and tertiary attention.Improve the existing coordination mechanisms between the three levels of primary, secondary and tertiary care.Use the methodology of pre-classification or pre-Triage of the patients [51] that reinforce the Triage, where it is possible to separate the demand according to the given classification where the patient is admitted.Design the attention flow and the management of spaces for the different kinds of attention, so that a dental consultant patient (C5) does not meet a C2 or C3 patient. This is as long as there are conditions in the physical plant and a prudent number of qualified professionals.Develop telemedicine [53,54,55,56] in some specialties, to facilitate a more efficient and faster attention flow in terms of immediate responses.Define the beginning of the waiting period, since this causes certain difficulties at the moment of assigning the attention boxes, and the priority of later attention.

As shown in Table 4, the improvements can be evaluated with impact and effectiveness indicators. In this way, it is possible to measure [57] if each one of them responds efficiently to the actual needs of the users in relation with the overuse of attention in the Emergency Services of Chile. Impact is understood as the effect on the level of attention of each one of the actors that are interrelated in the process; and efectiveness, as the relationship that exists between the achieved results and the resources used, in terms of the expected and desired effect.

Studying a health system is complex. It requires more education for the professionals and a greater collaboration that benefits all the patients that require fast and quality attention. Even though care in the Emergency Service is only part of the problem, as seen by both the internal and external users, it is necessary to observe it as a system. The design of the service must be meticulously studied, asking each user about their concerns, and with the information obtained it would be possible to promote an efficient service, more friendly and human.

In Figure 3, the results of the evaluation of Table 4 are shown. In the graph, it can be seen in the evaluation that the best evaluated proposal is “use of ICT” since it obtains 71.4%, followed by “Promotion of training in the professionals of the PHC”, “Promotion of emergency medicine training” and “Development of telemedicine in some specialties” with 57.1%. It is observed that “Pre-classification methodology” was the most heterogeneous measure in which 14.3% of the experts rate it as insufficient.

To execute the proposals, it is necessary to have the collaboration of the health personnel and the authorities in charge of the government through the intervention of the Ministry of Health, Hospital Directors, medical and administrative personnel, and patients. Therefore, it is essential that from the Undersecretary of Assistance Networks, the problems that occur in the Emergency Services be analyzed in order to collaboratively evaluate the feasibility of projects that promote improvements to the system, and thus guarantee access to urgent care in a timely manner, with a quality service for all.

On the other hand, there is a deficit of economic resources and a non adequate distribution of them. It is required to design the service starting from a multidisciplinary point of view, where not only providers of the service are included, but also architects, engineers of various disciplines, and anyone who could be a contribution for this to improve the system in the long term.

## 6. Limitations

Some limitations are observed for the execution of the improvement proposals:Lack of participation of the external user (patient) in the analysis and execution of the improvement proposals. In Chile, there is no formal Association of patients who participate in public health policies, which is also a relevant issue that limits such actions.Bureaucratic aspects in the decision-making process of the Chilean Health System tend to be long and difficult. They end when an initiative benefits a line of action.The SARS Cov-2 pandemic has restricted the public agenda related to health issues. The authorities of the Health System have refocused their functions and the allocation of resources towards the health emergency.There is a need for allocation of more economic resources to incorporate more technology, train workers, take actions that reduce resistance to change, and encourage innovation.There is a need to strengthen the education about health promotion of people.There is a need for more efficient and effective management, at the level of primary care in family medicine as well as in specialties, to contribute to a decisive, assertive and quality health care.There is a need for more access to the information obtained from records and statistics, which are generally confidential.

## 7. Conclusions

Studying the health emergency service in Chile is complex since there are multiple economic and social difficulties that affect each of its actors, which is why an action plan is required that provides solutions to the entire system. Some critical factors that need to be improved are observed, such as: scarce human resources, depersonalized attention, little effectiveness and efficiency in the management and handling of indicators, lack of investment in ICT and telemedicine, which allows faster and more organized attention, among others. It should be noted that the aforementioned factors are mainly related to the management of primary care and have a direct effect on the tertiary level.

According to the main causes that have been reported by both internal and external users, a significant amount of tertiary level patients saturate the Emergency Department because they have not had the opportunity to solve their health problems in the Primary Care Network due to difficulties in accessing medical appointments, and lack of time flexibility of the medical agenda. Additionally, it has been reported that anomaly situations in the patients condition are not correctly referred in time within the Assistance Network for analysis and treatment, which also overloads the Emergency Department. Another cause reported is the lack of education that people have about health promotion and prevention, which also contributes to the saturation of the Emergency system. Patients do not know how to choose the right level of care and the good use of the Assistance Network, in general terms.

As mentioned in the discussion, the problems of the health system need to be studied by multidisciplinary and interdisciplinary teams of professionals who should improve their social skills, such as empathy and emotional intelligence. In this way, health professionals can provide a more enjoyable, comprehensive and efficient care. It should be noted that it is difficult to promote these competencies since there are strenuous shifts with overuse of medical care by patients on a daily basis.

On the other hand, according to proposals highly valued by experts, it is necessary to promote the training of professionals both in Primary Care and in Emergency Medicine. Moreover, the amount of health professionals is another issue, evidenced by a study published by the OECD. The proportion of doctors and other health professionals in Chile between 2016 and 2018 is lower than the OECD average (on average there are 2.5 and 3.5 doctors per 1000 persons, respectively).

The proposals could be implemented if public policies increase the funding on emergency health in the public budget. If a diagnosis of the care process were carried out, then each professional could commit to use the new technologies and to improve their skills to provide fast and quality health service. With the contingency, it is very important to search for new mechanisms, opportunities and solutions that favor the improvement of the excessive use of medical care in the Emergency Health Service.

Finally, it is worth mentioning that the COVID-19 pandemic has revealed the difficulties and limitations of the health system, showing the deficiencies that exist at all levels of care, the availability of hospital beds, the points of care for critical patients, and the response capacity of Emergency Services. It is necessary to establish protocols, new forms of care, strengthen preventive health programs, and emphasize the training of health professionals in relation to this type of crisis and others. It is interesting to see how a pandemic is capable of placing evidence of the deficiencies of health services around the world [59]. Therefore, Chile has not been an exception, since this threat has become a cause for concern, affecting query waiting times.

## Figures and Tables

**Figure 1 ijerph-18-03082-f001:**
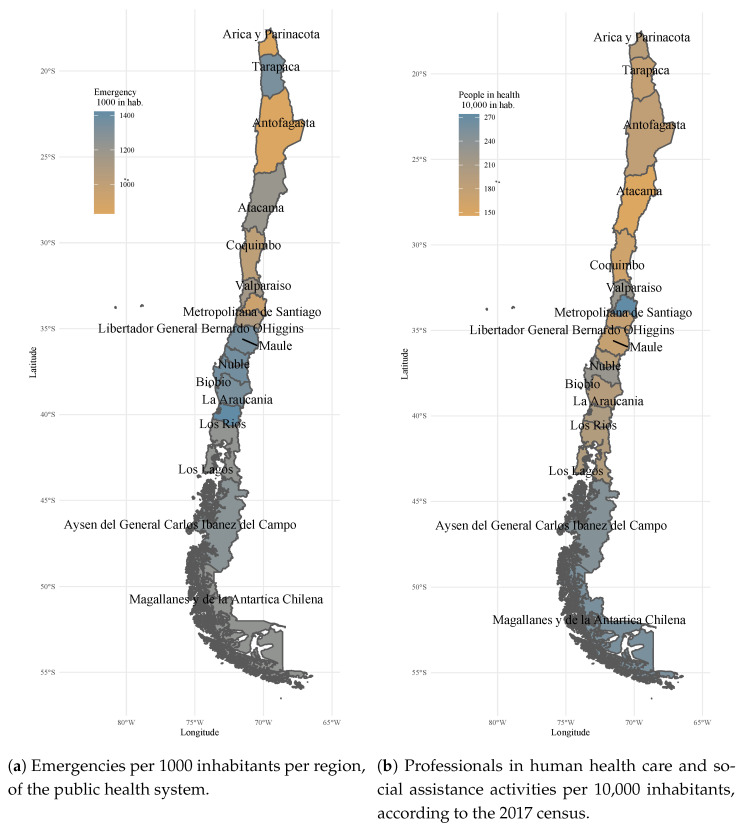
Emergencies and people in health care system.

**Figure 2 ijerph-18-03082-f002:**
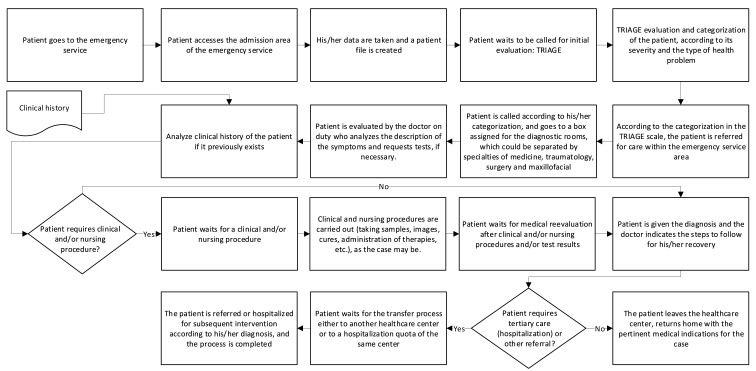
Activities in an average emergency care.

**Figure 3 ijerph-18-03082-f003:**
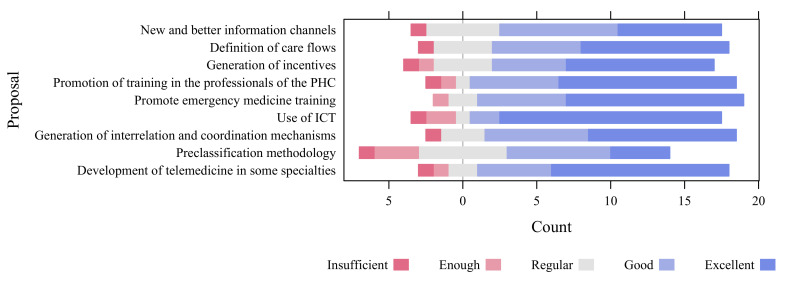
Valuation of proposals for improvements by experts in Emergency Medicine, with 21 responses [58].

**Table 1 ijerph-18-03082-t001:** Number of emergency, primary and specialty care attentions per Region in 2017 and population according to 2017 census.

Region	Emergency Care		Primary Care		Specialties Care		Total	Emergency	People in Health	Population
	Number	%	Number	%	Number	%	Care	1000 in hab.	10,000 in hab. ${}^{a}$	
Arica and Parinacota	188,358	38.1%	175,268	35.4%	130,961	26.5%	494,587	833	189	226,068
Tarapacá	442,356	48.6%	332,969	36.6%	134,111	14.7%	909,436	1338	177	330,558
Antofagasta	512,673	44.3%	387,985	33.5%	256,491	22.2%	1,157,149	844	182	607,534
Atacama	343,906	49.0%	230,483	32.9%	126,833	18.1%	701,222	1202	146	286,168
Coquimbo	768,004	43.8%	639,912	36.5%	346,521	19.8%	1,754,437	1014	164	757,586
Valparaíso	2,092,230	46.9%	1,641,461	36.8%	724,375	16.2%	4,458,066	1152	226	1,815,902
Metropolitan of Santiago	6,666,899	40.9%	6,445,116	39.5%	3,203,522	19.6%	16,315,537	937	274	7,112,808
Libertador General Bernardo O’Higgins	993,010	42.3%	980,250	41.7%	375,563	16.0%	2,348,823	1086	177	914,555
Maule	1,407,230	46.5%	1,096,038	36.2%	523,404	17.3%	3,026,672	1347	175	1,044,950
Biobío and Ñuble	2,778,550	43.4%	2,482,816	38.7%	1,146,719	17.9%	6,408,085	1364	219	2037,414
La Araucanía	1,281,165	44.4%	1,168,756	40.5%	437,066	15.1%	2,886,987	1338	198	957,224
Los Ríos	547,319	47.5%	393,777	34.2%	211,553	18.4%	1,152,649	1422	204	384,837
Los Lagos	1,022,731	46.1%	786,680	35.4%	410,217	18.5%	2,219,628	1234	198	828,708
Aysén of general Carlos lbáñez del Campo	131,300	38.1%	126,528	36.7%	86,929	25.2%	344,757	1273	247	103,158
Magallanes and Chilean Antartic	206,326	45.6%	121,621	26.9%	124,699	27.5%	452,646	1239	256	166,533
Total country	19,382,057	43.4%	17,009,660	38.1%	8,238,964	18.5%	44,630,681	1103	229	17,574,003

${}^{a}$ Human health care and social assistance activities, census 2017.

**Table 2 ijerph-18-03082-t002:** Description of the Triage categorization method.

Category	Description of Seriousness
C1	Patients in vital risk, i.e., those that require reanimation and/or immediate stabilization, because of the imminent vital risk. They have direct priority access to the reanimation box, and their attention must be immediate. Whoever recognizes the patient emergency activates the emergency alarm.
C2	High complexity patients that require diagnostic and/or therapeutic actions, such as an evaluation, treatment and control for a period of time, which may require hospitalization and/or specialists’ consultation. Its most frequent characteristic is hemodynamics instability. The patient goes quickly to the box and the waiting time must not be more than 10 min.
C3	Medium complexity patients that due to the nature of their pathology require diagnostic-therapeutic measures to determine a brief period of observation and subsequent discharge. The patient goes to the medical care box, where they are evaluated by the doctor according to availability. The waiting time should not be more than 60 min.
C4	Patients with no real emergency. They are patients requiring a diagnostic procedure or a therapeutical one, including medical attention.
C5	General consultation, i.e., any clinical situation that appears spontaneously and/or for a long-term that can produce only general discomfort in the patient. Because of the associated clinical condition, both the medical attention as well as the initial indication of treatment can be solved through the Primary Health Care (PHC).

**Table 3 ijerph-18-03082-t003:** Average waiting times for patients attending an Emergency Department (Average time in minutes).

TRIAGE	TRIAGE Categorization	Medical Evaluation	Clinical Procedures	Medical Reassessment	Referral, Transfer or Hospitalization	Total
C1	As these are life-threatened patients, waiting times from C2 to C5 are not generated					
C2	18.1	22.2	32.5	38.2	37.5	148.5
C3	32.3	48.5	70.4	49.6	188.7	389.5
C4	40.5	90.3	75.5	128.4	52.7	387.4
C5	39.2	101.9	98.3	196.8	64.3	500.5

**Table 4 ijerph-18-03082-t004:** Proposal, Description, Measurement and Observations of the detected opportunities.

Proposal	Causes of the Problem of Interest	Description of the Solution	Measurement	Observations
(1) New and better information channels	Poor coordination between the patients and the health providers. Little education and promotion in relation to the preventive health in the patients and the good use of the service in general terms. Lack of unification of the service at PHC and differences among administrations.	Generation of new information channels making the distinction of the existence of a particular channel for the health center, the patients, and between them. Improve the connection of the information channels with the external user with the aim of guiding them in relation to the corresponding healthcare network, and improving the management of the reservation of medical appointments.	Impact	Patients would find themselves informed about the health centers and about the management of the reservation of medical appointments.
(2) Definition of medical care flows	Hospitals have legal obligations according to law, causing saturation of the system.	Definition of care flows for the different category of patients, such as a fast tracking or area for quick attention of C4 and C5 patients, leaving the observation areas only for the patients that can develop changes in their health state, such as C2 and C3 patients that have more probabilities of worsening or improving their condition. Management of the medium severity and not serious patients (C3-C4) demand can reduce the waiting time that can put in danger the care or the resources intended for the serious ill patients; as well as improve the perception of the service.	Effectiveness	It would considerably improve medical care flows. It requires the reinforcement of the medical staff and the infrastructure of the place.
(3) Generation of incentives	Lack of incentive for general practitioners and health professionals who are part of the PHC structure. Few economic resources earmarked for the hiring of more professionals for the public health network in the three levels of attention.	Generate an incentive in the health professionals so that the bet is the contribution of the public health service. To obtain more professionals that can give responses to the population.	Impact/ Effectiveness	The increase of the budget is required for public health.
(4) Promotion of training in the professionals of the PHC	Patients have limited access to the PHC. Little education and promotion in relation to the preventive health in the patients and the good use of the service in general terms. Lack of incentives for general practitioners and health professionals who are part of the PHC structure.	Promotion of the training of decision-making skills in the primary attention professionals, through the specialization in family medicine so they can give an accurate diagnosis, in order to educate the population in relation to their health condition.	Efectiveness	The doctor must have received professional training, and later made an effort through some type of integral training, diplomas and/or Master’s among others to favour this opportunity.
(5) Promote emergency medicine training	Few economic resources earmarked for the hiring of more professionals for the public health network in the three levels of attention.	Promotion of emergency medicine training for those who dedicate themselves to this type of attention, so they can sharpen their clinical eye on the immediate response.	Effectiveness	The doctor must have received professional training, and later make an effort through some type of integral training, diplomas and/or Master’s among others to favour this opportunity.
(6) Use of ICT	There is not always a medical record of the patients who go to the Emergency Services. Greater investments are required for Health Technology Assessment to improve centers.	Use of ICT to create systems allowing the access to the medical history of the patients, making this information available in any type of facility where the patient has arrived to ask for care. Creation of an electronic medical record system containing the patient medical history [57], and thus, favour the communication mechanisms among the various health centers at the time of provision of the medical care.	Impact/ Effectiveness	It will allow a prompt medical care through accessing the clinical history of the person attended, helping the diagnosis be faster and more accurate.
(7) Generation of interrelation and coordination mechanisms	Poor coordination between the patients and the health providers.	Generation of interrelations and coordination mechanisms among the three levels of care: primary, secondary and tertiary.	Impact	It will favor the communication and will be a more effective provision of services among all the actors.
(8) Pre-classification methodology	Hospitals have legal obligations according to law, causing saturation of the system.	Use of a pre-classification methodology of patients that reinforces the Triage [51], to separate the demand according to the classification given at the moment in which the patient is admitted. Designing care flows and management of the spaces for the different types of attentions required by C5 patients, or for a dental consultation, so that they do not run into a C2 or C3 patient attention sector (as long as there are conditions of the physical plant and a prudent amount of qualified professionals).	Effectiveness	It could be effective as long as the demand of staff is reinforced. On the contrary, it would be difficult for the simple fact that the experience is what determines, most of the time, the emergency that the patient is facing.
(9) Development of telemedicine in some specialties	Patients have limited Access to the PHC. Little education and promotion in relation to the preventive health in the patients and the good use of the service in general terms. Greater investments are required for Health Technology Assessment to improve medical centers.	Promotion of the use and development of telemedicine in some medical specialties to favor a more efficient and faster care flow in terms of immediate responses.	Impact	Its use will help various specialties which can be cared for from distance so it does not generate an increased flow of patients. In addition, this will allow the generation of a follow-up of the condition of the more immediate patient.

## Data Availability

Not applicable.

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
