# Peer review of "Overuse of Health Care in the Emergency Services in Chile"

_ijerph, 2021, doi:10.3390/ijerph18063082_

Round 1

Reviewer 1 Report

Dear Authors

Greetings

Thank you for making text performance optimizations. Congrats! Thank you for your consideration, fully response and efforts. I included the attached doc with some last suggestions.

Best regards

Author Response

Greetings

Thank you for making text performance optimizations. Congrats! Thank you for your consideration, fully response and efforts. I included the attached doc with some last suggestions.

Author response: We thank the reviewer for his comments and suggestions that have helped us to improve our work.

Author action: The suggestions indicated have been considered in the new version of the manuscript.

Reviewer 2 Report

accept

Author Response

accept

Author response: We thank the reviewer for his comments and suggestions that have helped us to improve our work.

Author actions: —

Reviewer 3 Report

Revision of the article (second revision):

Overuse of care in the emergency services in Chile

The subject of the article is very interesting and the article provides many important ideas for improving care in emergency services and to advance in the sustainability of the public health service. It also provides a good reflection on the use of the emergency service and the need to make a thorough review of the use of the rest of the levels of care.

Despite acknowledging the improvements made in the writing of the article, it still lacks important aspects of an original article according to the instructions of the journal.

The main areas for improvement detected are explained below:

Author List and Affiliations: These data should be written in English.

Abstract: The abstract is not informative and does not collect or clearly explain the objective of the article, the design, the main results and the most relevant conclusions.

Introduction: The introduction is very long in relation to the rest of the sections of the article. In addition, sections 2 "The health service user" and 3 "Description of the attention process" contain information that should also be part of the introduction.

Materials and Methods: The article lacks this section. The article still doesn´t sound scientifically rigorous. It is well written, with a very clear wording, but it is not specified what type of article it is, nor what the design is, nor how the improvement proposals have been prepared...

Results: The article lacks this section.

Discussion: The improvement proposals included in the discussion section are very interesting to improve care in emergency services, but they should be included and explain the design used to develop them (research process, literature review, panel of experts …) in the results section.

Section 5COVID 10 Situation”  is without connecting thread with the rest of the article

Table 3 contains very relevant information, but it should be contextualized in a clear objective and in a clear design and development, explaining everything well beforehand in the article.

Conclusions: The conclusions are not known from where they derive. They are very generic and not based on hard data.

References: As I pointed out in the first review of the article, the References section needs to review some aspects according to the journal's referencing standards, such as referencing the name of the journal in abbreviated form. You can see the instructions to reference the sources of information used on the web: https://www.mdpi.com/journal/ijerph/instructions

Kind regards

Author Response

Author response: We greatly appreciate comments on the areas for improvement detected, for this reason we have applied improvements in the manuscript in the areas indicated by the reviewer.

Author actions: Each of the improvement actions is listed below:

* Author List and Affiliations: This information has been written in Spanish since it is the protocol

indicated by most of the universities in Chile.

* Abstract: The abstract has been improved to be informative, explaining the article’s objective, the design, results and the most relevant conclusions.

* Introduction: The introduction has been reformulated and reduced. Additionally, a "Background" section has been added, including representative tables and figures.

* Materials and Methods: Section 3 has been improved based on feedback from reviewers. In general terms, the section 3 includes "Materials and Methods".

* Results: Section 4 has been improved based on feedback from reviewers. In general terms, the section 4 includes "Results and Discussion".

* Discussion: This section has been improved based on feedback from reviewers.

* Conclusions: This section has been improved based on feedback from reviewers.

* Section 5 “COVID 10 Situation”: This session has been eliminated and its content has been distributed in different sections to improve the context of the manuscript.

* Table 3: Table 3 has been improved, contextualized, and renumbered (current Table 4).

* References: This section has been improved based on feedback from reviewers. The manuscript has been generated with TEX, including the libraries indicated by MDPI and following the standard MDPI format.

We thank the reviewer for his comments and suggestions that have helped us to improve our work.

Reviewer 4 Report

The authors have answered most of the questions raised in the review.
Although some of the suggested changes have not been incorporated into
the text, I think the revised document has improved over the original
version.

Author Response

Author response: We thank the reviewer for his comments and suggestions that will help us improve our work.

Author actions: We have re-reviewed the previously proposed questions (round 1 review) and have included the responses to the reviewer’s questions and concerns in the current manuscript. As well as his suggestions.

Round 2

Reviewer 3 Report

Revision of the article (third revision):

Overuse of care in the emergency services in Chile

Some aspects indicated in the previous review have been improved and resolved, but the following aspects remain unchanged:

  • Materials and Methods: It describes how the care process is in the emergency unit, but information on the materials and methods section of this article is still lacking. It is not specified what type of article it is, nor what the design is, nor how the improvement proposals have been prepared...
  • Results: Some information included in the results section should be included in the materials and methods section. For example, how the proposals in Table 4 were evaluated with surveys of 21 experts in Emergency Medicine, etc.
  • Conclusions: In the conclusions section, many opinions are issued, without being based on the data provided in the previous section of results. In addition there are inappropriate and insufficiently argued phrases such as "An important part of the tertiary level patients saturate the Emergency Department, this happens because users have not had the opportunity to solve their health problems in the Primary Care Network and / or due to the lack of education that people have in this area of health promotion and prevention, it is difficult for them to choose what level of care corresponds to them according to the type of ailment they have”.

I hope these appreciations are welcome and help to improve the article.

Kind regards

Author Response

This manuscript is a resubmission of an earlier submission. The following is a list of the peer review reports and author responses from that submission.

Round 1

Reviewer 1 Report

Dear Authors

Greetings

Congratulations for your research. I would like to suggest one review about the presentation design of tables and figure 1. Maybe to consider to present some graphics or other presentation resources. In addition, please find additional comments in the attachment.

Best regards

Reviewer 2 Report

Thank you for the opportunity to review the manuscript. The article is interesting and deals with the important problem of abuse of emergency services. However, I have a few comments to the authors and please take them into account:

1) I believe that the style / language of the work could be better

2) stylistically and grammatically incorrect sentences appear, e.g. ". Starting from that, we can ask the following questions: Are all patients who arrive at the emergency service really a medical emergency ?; and Who defines the emergency, the user or the system ?. " - it definitely requires linguistic changes.

3) The concept of "emergency service" should be differentiated as the authors use it for both emergency units
(Emergency Medical Service) and hospital units (Emergency Department). Maybe these are the authors' language problems?

4) The discussion is definitely too short. I propose to transfer some of the content from the APPLICATIONS to the DISCUSSION.

5) There is no clear indication by the authors of the LIMITATIONS OF THE WORK.

6) The authors mention mainly hospital and outpatient care. They point to TRIAGE as an effective form of verification of patients' needs. There is no description of the verification of reports from EMS units and their procedures at the scene of the incident to verify the needs of patients.

7) During a pandemic, the work becomes outdated, because it concerns the system from 2017. It is obligatory to supplement the article with changes that occurred during the pandemic.

6) The work contains 42 items of literature, but as many as 18 of them (43%) are articles older than 3 years. I recommend that you update your references with the following articles:

a) Mitura, K. The impact of COVID-19 pandemic on critical care and surgical services availability. Crit. Care Innov. 2020, 3 (2), 43-50.
DOI: 10.32114 / CCI.2020.3.2.43.50
(This article describes the impact of an epidemic on in-hospital procedures for critical patients. Care at the Emergency Department varies significantly today.);

b) Curt, N .; Tintet, L .; Chavada, P .; Mazet, G .; Combes, D .; Dekesel, B. "112: CAN I HELP YOU?" - an european first aid education project. Crit. Care Innov. 2020, 3 (1), 9-17.
DOI: 10.32114 / CCI.2020.3.1.9.17
(The article gives an example of an international educational project for the society. The authors should provide such examples of system solutions in their work.)

c) Leszczyński, P .; Charuta, A .; Łaziuk, B .; Gałązkowski, R .; Wejnarski, A .; Roszak, M .; Kołodziejczak, B. Multimedia and interactivity in distance learning of resuscitation guidelines: a randomized controlled trial. Interactive Learning Environments, 2018, 26 (2), 151-162.
DOI: 10.1080 / 10494820.2017.1337035

d) Leszczyński, P .; Gotlib, J .; Kopański, Z .; Wejnarski, A .; Świeżewski, S .; Gałązkowski, R. Analysis of Web-based learning methods in emergency medicine: randomized controlled trial. Archives of medical science: AMS, 2018, 14 (3), 687.
DOI: 10.5114 / aoms.2015.56422
(the two articles above show how you can educate emergency medical professionals in order to improve their quality of health services)

e) Glue, T .; Jayamaha, AR. Knowledge of the in-hospital resuscitation algorithm among medical staff of selected hospital departments. Crit. Care Innov. 2019, 2 (2), 9-16.
DOI: 10.32114 / CCI.2019.2.2.9.16
(the study shows a gaps in the knowledge of medical staff in the hospital)

f) Kożuchowski, S .; Wejnarski, A .; St Cyr, A. The role of the Helicopter Emergency Medical Service as support for ground emergency medical services. Crit. Care Innov. 2019, 2 (3), 19-26.
DOI: 10.32114 / CCI.2019.2.3.19.26
(description of the characteristics of air ambulance activities)

Reviewer 3 Report

Revision of the article:

Overuse of care in the emergency services in Chile

This article addresses the organization, difficulties and opportunities for improvement in emergency services in Chile. The objective and the methodological design followed for the present study is not sufficiently explicit. The article doesn't sound scientifically rigorous. Therefore, it needs a thorough review prior to assessing its possible publication in this journal. It would also be necessary to review the references section and correct some aspects such as the name of the journals that should go with the abbreviation.

Kind regards

Reviewer 4 Report

The study reviews some of the main problems in the functioning of the Chilean health system and makes proposals for improvement, focused on Emergency Services.In my opinion, it is an interesting topic generalizable to other countries. Furthermore, the analysis made by the authors has practical implications that can improve healthcare system care.

However, I would like the authors to expand on some information that is included in the study.

  • Could the authors explain how they identified the 9 problems included in section 4? In addition to what is stated in the manuscript, has a previous study of needs analysis been carried out in patients and professionals? If it has not been done, do you think it would be useful?
  • It is not clear on what the authors have based to make the list of proposals in table 3. I think it's important to clarify. Has it been their own knowledge of the subject that has led them to identify these proposals? What professionals have intervened to make this list?
  • It would be interesting for the authors to indicate, if possible, whether the proposals have been applied in other countries and whether their effectiveness has been monitored.
  • Could the authors relate the proposals in table 3 with the problems identified in section 4?
  • It would be pertinent to state who would be responsible for carrying out the solution proposals in Table 3 (institutions, type of professionals) and assess the resources necessary to implement them and their viability. All proposals have the same relevance? All proposals have the same viability?
  • A section on limitations should be included. For example: not all lines of action are included, need to carry out a needs analysis of patients and professionals...

In summary, I think the paper is interesting, but it is necessary to provide more information on the issues outlined above.